# Salvianolic Acid B Regulates Oxidative Stress, Autophagy and Apoptosis against Cyclophosphamide-Induced Hepatic Injury in Nile Tilapia (*Oreochromis niloticus*)

**DOI:** 10.3390/ani13030341

**Published:** 2023-01-18

**Authors:** Liping Cao, Guojun Yin, Jinliang Du, Rui Jia, Jiancao Gao, Nailin Shao, Quanjie Li, Haojun Zhu, Yao Zheng, Zhijuan Nie, Weidong Ding, Gangchun Xu

**Affiliations:** Key Laboratory of Integrated Rice-Fish Farming Ecology, Ministry of Agriculture and Rural Affairs, Freshwater Fisheries Research Center, Chinese Academy of Fishery Sciences, Wuxi 214081, China

**Keywords:** salvianolic acid B, oxidative stress, *Oreochromis niloticus*, apoptosis, autophagy

## Abstract

**Simple Summary:**

Nile tilapia (*Oreochromis niloticus*) as one of the main commercially cultivated fishes in China, is plagued by liver diseases caused by some stressors. Salvianolic acid B (Sal B) has the strongest antioxidant effect and has a good protective effect on the liver. The objective of this study is to evaluate the protective effect of Sal B on oxidative hepatic injury in Nile tilapia, followed by assessing the potential protective mechanism. Our experimental data show that Sal B can not only directly reduce the level of reactive oxygen species (ROS), but can also resist CTX-induced oxidative injury by adjusting the autophagy and apoptosis levels of hepatocytes. Notably, the hepatoprotective effect of Sal B on Nile tilapia is related to the regulation of Nrf2, AMPK/mTOR and MAPK pathways. The research will lay the foundation for the application of Sal B in tilapia culture.

**Abstract:**

Salvianolic acid B (Sal B), as one of the main water-soluble components of *Salvia miltiorrhizae*, has significant pharmacological activities, including antioxidant, free radical elimination and biofilm protection actions. However, the protective effect of Sal B on Nile tilapia and the underlying mechanism are rarely reported. Therefore, the aim of this study was to evaluate the effects of Sal B on antioxidant stress, apoptosis and autophagy in Nile tilapia liver. In this experiment, Nile tilapia were fed diets containing sal B (0.25, 0.50 and 0.75 g·kg^−1^) for 60 days, and then the oxidative hepatic injury of the tilapia was induced via intrapleural injection of 50 g·kg^−1^ cyclophosphamide (CTX) three times. After the final exposure to CTX, the Nile tilapia were weighed and blood and liver samples were collected for the detection of growth and biochemical indicators, pathological observations and TUNEL detection, as well as the determination of mRNA expression levels. The results showed that after the CTX treatment, the liver was severely damaged, the antioxidant capacity of the Nile tilapia was significantly decreased and the hepatocyte autophagy and apoptosis levels were significantly increased. Meanwhile, dietary Sal B can not only significantly improve the growth performance of tilapia and effectively reduce CTX-induced liver morphological lesions, but can also alleviate CTX-induced hepatocyte autophagy and apoptosis. In addition, Sal B also significantly regulated the expression of genes related to antioxidative stress, autophagy and apoptosis pathways. This suggested that the hepatoprotective effect of Sal B may be achieved through various pathways, including scavenging free radicals and inhibiting hepatocyte apoptosis and autophagy.

## 1. Introduction

As an important metabolism organ, the liver can widely participate in the metabolism of inorganic salts, vitamins and other nutrients. At the same time, as an detoxification organ, it can decompose or transform various drugs, poisons and metabolites inside and outside the body into non-toxic substances and discharge them out of the body, playing an important role in maintaining the health of the body. In recent years, with the continuous aquaculture intensification, nutrient deficiencies, antibiotic abuse and drug residues have become prevalent, and all of these undesirable factors can lead to hepatic injury in aquatic animals [1,2]. Liver injury or lesions often lead to metabolic disorders and reduced disease resistance in fish, which can easily lead to outbreaks of secondary infectious diseases and can seriously restrict the healthy development of aquaculture [3,4,5]. At present, various antibiotics, pesticides and antiviral drugs cannot effectively control the occurrence of fish liver disease [6], which has become one of the factors affecting the development of aquaculture. In recent years, a large number of studies have shown that traditional Chinese medicine (TCM) can effectively promote growth and enhance immunity and disease resistance [7,8], and has good effects and unique advantages in combating fish liver injury [9,10]. The research into fishery production practices has proven that TCM as a feed mixer or feed additive method was particularly suitable for the current intensive and large-scale production of aquaculture, and was fully compatible with the disease control guidelines for the development of pollution-free aquaculture and the production of green aquatic products [11].

Salvianolic acid B (Sal B), isolated from the roots of *Salvia miltiorrhizae bunge* of Labiat, is the most abundant and active member of the total salvianolic acid [12]. Sal B, as one of the natural products with the strongest antioxidant effect [13,14], has a good protective effect in the treatment of cardiovascular diseases and liver fibrosis [15,16,17,18], which is inseparable from its strong ability to eliminate free radicals, reduce lipid peroxidation and regulate the antioxidant defense system. Through clinical observations and liver biopsies before and after treatment, Liu et al. [19] confirmed that Sal B can play an antihepatic fibrosis role by inhibiting the activation of hepatic stellate cells (HSCs) and reducing the deposition of collagen fibers. Wang et al. [20] and Li et al. [21] also showed that high concentrations of Sal B could inhibit the proliferation of Hep G2 cells and promote their apoptosis. At present, the hepatoprotective mechanism of Sal B is not fully understood, and there are few related studies in aquatic animals.

Cyclophosphamide (CTX) is a broad-spectrum antitumour drug and one of the most widely used immunosuppressive agents [22]. It has been widely used in aquatic products to build immunosuppressive models to study the immunomodulatory effects of immunopotentiators [23,24]. However, in the course of its clinical application, it has toxic reactions to the liver due to its ability to convert into cytotoxic alkylates in vivo [25], resulting in oxidative hepatic injury [26,27,28,29,30], which is often used to establish liver injury models to evaluate the hepatic protection of drugs. Xu et al. [31] explored the protective effect and mechanism of myricetin by using CTX to induce liver injury in mice. Fu et al. [32] found that polysaccharides of *Atractylodes macrocephala koidz* had a significant protective effect on CTX-induced liver injury in chicks. Pan et al. [33] also proved that drinking a ginseng stem-leaf saponin solution can alleviate oxidative stress induced by CTX in mice. However, in aquatic animals, there are few studies on the mechanism of liver injury induced by CTX. The further elucidation of the molecular mechanism of CTX-induced hepatic injury in fish is important for the subsequent screening and development of liver-protective drugs. Nile tilapia (*Oreochromis niloticus*) has the advantages of strong adaptability, high yield and fast growth rates. It is one of the dominant breeding species prioritized by the Ministry of Agriculture in China. The breeding quantity and output are huge, showing broad breeding prospects. In this study, CTX was used as a hepatotoxic agent to induce oxidative hepatic injury in Nile tilapia. The effects of Sal B on the hepatic antioxidant function, autophagy and apoptosis in Nile tilapia were evaluated, and the hepatoprotective mechanism of Sal B was preliminarily described, which laid the foundation for the application of Sal B in tilapia culture.

## 2. Materials and Methods

### 2.1. Animals and Treatments

The Nile tilapia (mean body mass (30 ± 5) g) were provided by the Freshwater Fisheries Research Centre of the Chinese Academy of Fisheries Sciences and domesticated in a recirculating water culture system for 2 weeks. The tank volume was 250 L, the water temperature was 26 ± 2 °C, the pH range was 6.8–7.5 and the dissolved oxygen rate was >6.0 mg·L^−1^. During domestication, the tilapia were fed a basal diet (Tongwei, Chengdu, China) at 4% body mass twice daily. During the test, we continued to the fill the oxygen, sucked dirt once every 2 days and replaced 1/5 of the clean water. At the beginning of the experiment, 300 tilapia were randomly divided into 5 groups, with 3 replicates per group and 20 fish per replicate. There was no significant difference in initial body weight between the replicates (*p* > 0.05), according to previous reports and our preliminary experiments [34,35]. The 5 groups were as follows: blank control group (Ctrl), model control group (CTX), low-dose Sal B (Sigma, St. Louis, MO, USA) group (0.25 g·kg^−1^ of Sal B), medium-dose Sal B group (0.50 g·kg^−1^ of Sal B) and high-dose Sal B group (0.75 g·kg^−1^ of Sal B). The formation of the experimental diets is outlined in Table 1, in which the tilapia in the Ctrl and CTX groups were fed the basal diet and fish in the Sal B-treated groups were fed the diets containing 0.25, 0.50 and 0.75 g·kg^−1^ of Sal B, respectively. The fish were fed twice a day at 4% of body weight, and the amount of bait was recorded and adjusted according to the feeding and growth conditions for a total of 60 days. Then, the tilapia in the CTX and Sal B-treated groups were given a intrapleural injection of 50 mg·kg^−1^ of CTX (Sigma, St. Louis, MO, USA), while the Ctrl group was given the same amount of normal saline. The injection was repeated 3 times every 3 days. The fish were fasted after injection [36]. The mortality rate of the tilapia was 0 during the experiment. Then, the Nile tilapia were anesthetized with 100 mg·L^−1^ of MS-222 (Sigma, St. Louis, MO, USA) [4] and weighed. Eight fish were randomly selected from each group to collect blood and liver tissues. The serum was separated from the blood via centrifugation at 3000 r·min^−1^ for 15 min at 4 °C and then stored at −80 °C. The animal experiment was carried out considering the animals’ welfare, and was approved by the Ethics Committee of the Chinese Academy of Fishery.

### 2.2. Determination of Growth Indicators

The relative weight rate (RWR, %), specific growth rate (SGR, %·D^−1^) and feed conversion ratio (FCR) of each group were calculated according to the following formula. RWR = (*W_t_* − *W*_0_)/*W*_0_ × 100%; SGR = (ln*W_t_* − ln*W*_0_)/t × 100%; FCR = F/(*W_t_* − *W*_0_). In the formula, *W_t_* and *W*_0_ represent the final and initial body weights (g) of the Nile tilapia, respectively; t stands for the feeding time (d); F stands for the feeding amount (g).

### 2.3. Histopathological Observations

Here, 5 mm^3^ of liver tissue was cut with forceps, fixed in 4% paraformaldehyde for more than 24 h, dehydrated in 50–100% ethanol solution, made transparent with xylene and embedded in paraffin at 56–58 °C. Then, the paraffin-embedded blocks were sectioned to 4–5 μm in thickness, stained with H&E (hematoxylin and eosin) and finally sealed with optical resin on slices for microscopic observations.

### 2.4. TUNEL Assay

Hepatocyte apoptosis in Nile tilapia was detected via terminal deoxynucleotidyl transferase-mediated dUTP nick end labeling (TUNEL) in paraffin sections, according to the Roche Biotech TUNEL kit (Roche, Switzerland). The paraffin liver sections were dewaxed twice in xylene, hydrated in gradient ethanol and incubated with 20 μg·mL^−1^ of proteinase K solution for 1 min at 37 °C. Endogenous peroxidase was then inactivated via the dropwise addition of 0.3% H_2_O_2_, followed by incubation with the freshly prepared TUNEL mixture at 37 °C under light-proof conditions. Finally, 3,3-diaminobenzidine (DAB) was added sequentially, followed by hematoxylin staining to return the color to blue and neutral gum to seal the film. A light microscope (DM750, Leica, Wetzlar, Germany) was used to observe the analysis and take the films.

### 2.5. Determination of Biochemical Indicators

The determination of relevant biochemical indicators in serum and liver samples was carried out using commercial kits. Serum indicators including alanine aminotransferase (GPT), aspartate aminotransferase (GOT), the total antioxidant capacity (T-AOC), total protein (TP) and albumin (Alb) [37]; and liver indicators including superoxide dismutase (SOD), catalase (CAT) and reduced glutathione (GSH) were measured using kits from Nanjing Jiancheng Bioengineering Institute Technology Co., Ltd. (Nanjing, China) [38]. Malondialdehyde (MDA) [39] and nitric oxide (NO) in the liver were determined according to the kits from Biyuntian Biological Company (Shanghai, China) [40].

The content of 8-OHdG in the liver was determined using an enzyme-linked immunosorbent assay (ELISA), according to the ELISA kit (Mlbio, Shanghai, China). In summary, 50 μL of sample and 100 μL of horseradish peroxidase (HRP)-labeled antibody were added successively after coating the microplate with a purified fish antibody. After incubation for 60 min at 37 °C, the chromogenic agent was added and the absorbance was read at 450 nm.

### 2.6. Determination of mRNA Level

The total RNA was extracted from Nile tilapia liver samples using the Trizol method. After 0.1 g of liver sample had been thoroughly ground in liquid nitrogen, the total RNA was extracted according to the instructions for the RNAiso Plus (Takara, Dalian, China # 9109). After electrophoresis was performed to determine the RNA integrity, the OD260/OD280 values (1.8–2.0) of the RNA were determined and the concentration was calculated. An appropriate amount of RNA was reverse-transcribed to synthesize cDNA according to the Prime Script™ RT reagent kit (Takara, # RR047A) and stored at −80 °C for later use.

Here, *q*PCR was performed using the SYBR Green fluorescent dye method, following the instructions of the ExScript™ RT-PCR kit (Takara, Dalian, China). The total reaction system was 25 μL/12.5 μL of TB Green Premix Ex TaqII (Takara, # RR820A), 0.5 μL of upstream and downstream primers, 9.5 μL of RNase-free water and 2 μL of cDNA. The reaction process included 40 cycles, with each cycle proceeding as follows: 95 °C for 30 s; 95 °C for 5 s, 60 °C for 30 s and 72 °C for 30 s. The relative quantification of the mRNA levels was performed using *β*-actin as an internal reference and finally calculated using the 2^−ΔΔCt^ method [41]. The target-gene-specific primers are shown in Table 2.

### 2.7. Statistical Analysis

All data were expressed as means ± standard errors (x ± SEs), and a statistical analysis was performed using SPSS 22.0 software. A one-way analysis of variance (one-way ANOVA) combined with an LSD test was used to compare the differences in the data between groups. Here, *p* < 0.05 was considered significant.

## 3. Results

### 3.1. Effects of Sal B on Growth Performance of Nile Tilapia

The effect of Sal B at different dietary levels on growth indicators of Nile tilapia are shown in Table 3. After 60 days of feeding, dietary supplementation with Sal B significantly increased the final body weight, RWR and SGR (*p* < 0.05) and decreased the FCR (*p* < 0.05) compared to the Ctrl group. Moreover, the beneficial effect was in a dose-dependent manner.

### 3.2. Effects of Sal B on the Histopathology of Nile Tilapia

The histomorphological examination of the liver revealed that the hepatic structure of the Ctrl group showed integrity, hepatic lobules were clearly visible, the hepatocyte cords were permutated in an orderly manner and no obvious lesions were observed (Figure 1A). In the CTX group, hepatocyte enlargement, extensive vacuolar degeneration and plasma osteoporosis were observed. Meanwhile, the nuclei were laterally shifted, pyknotic and hyperchromatic (Figure 1B). The dietary supplementation of Sal B can effectively inhibit the hepatic damage induced by CTX. In the 0.25 g·kg^−1^ Sal B pretreatment, the liver sections showed fusion between hepatocytes, blurred cell outlines, solidified and laterally shifted nuclei and more vacuolization (Figure 1C). Pretreatment with 0.50 g·kg^−1^ of Sal B can significantly reduce the degree of hepatic injury, increase the number of hepatocytes, reduce the vacuolization and produce clearly visible hepatocyte cords (Figure 1D). In the 0.75 g·kg^−1^ Sal B group, the pathological examination of the sections showed that the damage was significantly alleviated and there was no obvious cytopathy (Figure 1E).

### 3.3. Effects of Sal B on TUNEL Assay

The results of the TUNEL assay for hepatocyte apoptosis showed that the nuclei of hepatocytes in the Ctrl group were light blue, and no apoptotic cells were found (Figure 2A). A large number of apoptotic cells were seen after the CTX treatment, and the nuclei were stained brown (Figure 2B). The dietary supplementation of Sal B can effectively inhibit CTX-induced hepatocyte apoptosis in a dose-dependent manner. Low-dose Sal B had no obvious effect on inhibiting hepatocyte apoptosis, and the degree of apoptosis was severe (Figure 2C). Sal B pretreatment at the dose of 0.5 g·kg^−1^ significantly inhibited hepatocyte apoptosis (Figure 2D), and only a small amount of hepatocyte apoptosis occurred when the dose of Sal B increased to 0.75 g·kg^−1^ (Figure 2E).

### 3.4. Effects of Sal B on Biochemical Indexes of Nile Tilapia

The effects of Sal B on serum biochemical indexes of Nile tilapia are shown in Table 4. Compared to the Ctrl group, the activities of the GPT and GOT were significantly increased after the CTX treatment (*p* < 0.05), while the levels of TP, Alb and T-AOC were decreased significantly (*p* < 0.05). The dietary supplementation of Sal B could effectively restore the levels of the above biochemical indicators in a dose-dependent way. Compared with the CTX group, the Sal B pretreatment at 0.75 g·kg^−1^ had a significant regulatory effect on all of the measured biochemical parameters (*p* < 0.05). Similary, the GOT and Alb levels were inhibited by the 0.50 g·kg^−1^ Sal B pretreatment (*p* < 0.05). The 0.25 g·kg^−1^ Sal B pretreatment only had an inhibition effect on the GOT activity (*p* < 0.05).

In the liver (Table 4), the CTX challenge significantly inhibited the antioxidant capacity and the levels of SOD, GSH, CAT and T-AOC were significantly decreased (*p* < 0.05), while the level of lipid peroxidation (MDA) and the degree of hepatocyte DNA damage (8-OHdG) were significantly increased (*p*< 0.05). Moreover, NO was decreased significantly (*p* < 0.05). Compared with the CTX-treated group, the 0.75 and 0.50 g·kg^−1^ Sal B pretreatments significantly increased the levels of SOD, GSH, CAT, T-AOC and NO, while inhibiting the contents of MDA and 8-OHdG (*p* < 0.05). The 0.25 g·kg^−1^ Sal B treatment only had a significant effect on the recovery of the liver T-AOC content (*p* < 0.05).

### 3.5. Effects of Sal B on mRNA Levels of Antioxidant-Related Genes in Nile Tilapia

The effects of Sal B on the expression of antioxidant-related genes in Nile tilapia liver are shown in Figure 3A. The mRNA levels of glutathione peroxidase 3 (*gpx3*), glutamate–cysteine ligase catalytic subunit (*gclc*), glutathione reducase (*gsr*), glutathione s-transfer α (*gstα*) and catalase (*cat*) in the liver were significantly decreased in the CTX-treated group compared with the Ctrl group (*p* < 0.05). The 0.75 and/or 0.50 g·kg^−1^ Sal B pretreatments significantly upregulated the mRNA levels of the above-mentioned antioxidant-related genes (*p* < 0.05).

To further investigate the antioxidant mechanism of Sal B, the mRNA levels of genes related to the Nrf2 signaling pathway were examined (Figure 3B). The CTX treatment significantly inhibited Nrf2-pathway-related genes, including *nrf2*, uncoupling protein 2 (*ucp2*) and heme oxygenase-1 (*ho-1*) (*p* < 0.05), while it upregulated the gene expression of an *nrf2* repressor, Kelch-like ECH-associated protein 1 (*keap1*) (*p* < 0.05). However, compared with the CTX group, the downregulation of these genes was significantly inhibited in the 0.75 g·kg^−1^ Sal B treatment (*p* < 0.05). In addition, significant upregulation was also found in *keap1* mRNA levels with 0.50 and 0.75 g·kg^−1^ of Sal B (*p* < 0.05).

### 3.6. Effects of Sal B on mRNA Levels of Autophagy-Related Genes in Nile Tilapia

The effects of Sal B on autophagy-related gene expression in liver samples are shown in Figure 4A,B. After the CTX treatment, the mRNA levels of autophagy-related genes including autophagy (*atg*)-3, -5, -7 and -13; microtubule-associated protein 1 light chain-3B (*lc3b*); and *beclin1* in the liver were significantly upregulated, while sequestosome 1 (*p62*) was significantly decreased (*p* < 0.05). However, the upregulation of *atg*-3, -5, -7 and -13; *lc3b*; and *beclin1* and the downregulation of *p62* were effectively alleviated in the pretreatment with 0.75 g·kg^−1^ of Sal B (*p* < 0.05). Additionally, the 0.25 and 0.50 g·kg^−1^ Sal B pretreatments also had a significant downregulation effect on the *atg5* expression (*p* < 0.05).

Similarly, in order to further explore the antiautophagy mechanism of Sal B, the mRNA levels of genes related to the AMPK/mTOR signaling pathway were examined in this study (Figure 4C,D). The results showed that the CTX treatment significantly induced the expression of MP-activated protein kinase (*ampk*), unc-51-like autophagy-activating kinase 1 (*ulk1*), mucolipin-1 (*mcoln-1*), lysosomal associated membrane protein 1 (*lamp1*), UV radiation resistance associated gene (*uvrag*) and transcription factor EB (*tfeb*), and significantly inhibited the gene expression of mammalian target of rapamycin (*mtor*) (*p* < 0.05). However, the Sal B pretreatment at a dose of 0.75 g·kg^−1^ reversed the changes in gene expression, as described above. Similarly, the 0.50 g·kg^−1^ Sal B pretreatment also significantly inhibited *mcoln-1* and *uvrag* gene expression (*p* < 0.05).

### 3.7. Effects of Sal B on mRNA Levels of Apoptosis-Related Genes in Nile Tilapia

The effects of Sal B on the expression of apoptosis-related genes in Nile tilapia liver samples are shown in Figure 5A,B. The CTX treatment caused a significant increase in the expression of apoptosis-related genes including cysteinyl-aspartate-specific proteinase (*caspase/cas*)*-3*, *-8* and *-9*; cytochrome (*cytc*); Bcl-2-associated X protein (*bax*); and protein 53 (*p53*) (*p* < 0.05), while B-cell lymphom/leukemia-2 (bal-2) decreased significantly (*p* < 0.05). The 0.75 and 0.50 g·kg^−1^ Sal B pretreatments effectively reversed the changes induced by CTX (*p* < 0.05).

In order to further explore the antiapoptotic mechanism of Sal B, the mRNA levels of mitogen-activated protein kinase (MAPK) signaling pathway-related genes including c-Jun N-terminal kinase1 (*jnk1*), *jnk2*, apoptosis signal-regulating kinase 1 (*ask1*), p38 MAPK (*p38*) and extracellular regulated protein kinase *(erk2)* were detected via *q*PCR (Figure 5C). The results showed that the CTX treatment significantly promoted the expression of these genes (*p* < 0.05), while the 0.75 and 0.50 g·kg^−1^ Sal B pretreatments significantly inhibited the upregulation of these genes (*p* < 0.05). Meanwhile, the 0.25 g·kg^−1^ Sal B pretreatment also downregulated the mRNA levels of *jnk1* and *erk2* (*p* < 0.05).

## 4. Discussion

Studies have shown that CTX greatly increases the content of ROS in the metabolic activation process after oxidation by P450 in the liver. This will trigger lipid peroxidation in the endoplasmic reticulum membrane system, and the intracellular antioxidant defense system scavenges free radicals through non-enzymatic reactions, resulting in the rapid depletion of GSH [44,45,46]. In this study, CTX administration caused abnormal levels of GOT, GPT, TP and Alb in serum samples and significant histopathologic changes. At the same time, the levels of T-AOC, SOD, CAT, NO and GSH in serum or liver samples also decreased significantly, and a large amount of MDA was produced. The content of 8OHdG, a marker of DNA oxidative damage, also increased significantly [47]. These results indicated that the damaged antioxidant defense system of Nile tilapia after CTX treatment resulted in a decrease in antioxidant capacity and an enhanced degree of lipid peroxidation in the liver, resulting in oxidative hepatic injury. However, the dietary supplementation of Sal B at doses of 0.75 and 0.50 g·kg^−1^ for 60 days was effective in suppressing changes of the biochemical indexes and effectively improve the pathologic changes in hepatic tissues. It was suggested that Sal B can increase the content of antioxidant enzymes by improving the biosynthesis and can inhibit lipid peroxygenation by scavenging free radicals, thereby antagonizing the oxidative damage caused by CTX and playing a protective role in the liver. In recent years, many experiments have confirmed that the antioxidant activity of Sal B plays a significant protective role in the process of liver injury either directly or indirectly. Tian et al. [12] found that the antioxidant activity and lipid peroxidation inhibition effects of Sal B were significantly stronger than those of Vc and rutin. The studies by Wang et al. [48] and Gao et al. [49] also showed that the antioxidant activity of Sal B has a significant therapeutic effect on CCl_4_-induced liver fibrosis in rats. Chen et al. [50] found that Sal B can reduce the lipid accumulation state and oxidative stress reaction of non-alcoholic fatty liver disease (NAFLD) cells, and plays a protective role in NAFLD cells.

Nrf2 is a key transcription factor regulating intracellular antioxidant stress responses [51], which can activate endogenous antioxidant responses and plays an important role in oxidative-stress-induced damage [52]. Studies have shown that the Nrf2/HO-1 antioxidant pathway plays an important role in Sal B, alleviating acute brain injury [53] and acute kidney injury [54] in rats and mice. In this study, the expression of Nrf2-pathway-related genes (except *keap1*) was inhibited by CTX, and the mRNA levels of its downstream phase II antioxidant enzymes (*ucp2*, *ho-1*, *gpx3*, *gclc*, *gsr*, *gstα* and *cat*) also decreased significantly. However, the 0.75 and 0.50 g·kg^−1^ Sal B pretreatments effectively inhibited the decline of these genes, indicating that Sal B also resisted CTX-induced oxidative hepatic damage in Nile tilapia through the Nrf2 pathway.

Cells degrade their own substances through autophagy, which increases the need for metabolic energy and the renewal of damaged organelles. Therefore, autophagy is of great significance for maintaining cell homeostasis and is a mechanism of self-regulation [55,56,57]. However, in pathological states, excessive autophagy exceeding its own lysosomal degradation capacity will affect normal cellular life activities, inducing apoptosis or lysosomal rupture and resulting in cell necrosis and accelerated disease development [58,59,60,61]. In recent years, there has been increasing experimental evidence that oxidative stress can induce autophagy [62,63,64], while the application of some antioxidants can also inhibit excessive autophagy and reduce cell death [65]. In the present study, the expression of the autophagy-related genes *atg-3*, *-5*, *-7* and *-13* was significantly upregulated after CTX treatment, and the expression levels of *lc3b* and *beclin1* as markers of autophagosomes [66] were also significantly increased. The expression of *p62*, a marker of autophagic degradation [67], was significantly decreased. However, the administration of 0.75 g·kg^−1^ of Sal B for 60 days significantly suppressed the changes. This result suggested that Sal B could reduce the intensity of the autophagy induced by CTX and alleviate liver injury. This result was consistent with the findings of He [68], who found that Sal B could exert a cerebral-protective effect on mice with acute focal cerebral ischemia by inhibiting the level of over-activated autophagy. Accordingly, we speculated that CTX inhibited the antioxidant capacity of the Nile tilapia, leading to the massive formation and accumulation of ROS, which not only led to oxidative hepatic injury in the tilapia, but also induced and enhanced the hepatocyte autophagy level. However, the strong free radical scavenging ability of Sal B played a certain role in inhibiting CTX-induced autophagy in the hepatocytes.

The AMPK/mTOR pathway is a classical pathway related to autophagy (Wang et al., 2020), and is widely involved in the pathologies of liver injury and liver disease [69,70]. AMPK and mTOR are two key factors regulating autophagy, which are involved in promoting and inhibiting autophagy, respectively. Meanwhile, AMPK negatively regulates mTOR [71]. Studies have shown that under stress conditions, activated AMPK phosphorylates mTOR to inhibit its activity, thereby activating the downstream ULK1 complex to induce autophagy [72,73,74]. In the present study, CTX promoted the initiation of autophagy by upregulating the expression of *ampk* and *ulk1* and inhibiting the mRNA level of *mtor*. Meanwhile, the regulatory genes (*mcoln-1*, *lamp1*, *uvrag* and *tfeb*) involved in lysosomal fusion during autophagy [75,76] were also significantly upregulated, which further strengthened the autophagy level of the hepatocytes. In contrast, the pretreatments with 0.75 and/or 0.50 g·kg^−1^ of Sal B significantly restored the *mtor* mRNA level and inhibited the upregulation of the other genes mentioned above. This result suggested that Sal B can inhibit the over-activated autophagy level, which may be related to its regulation of the AMPK/mTOR signaling pathway. This conclusion has been similarly reported in humans [77], mice [78] and rats [79].

Oxidative stress is an important factor affecting apoptosis [80]. The ROS generated in the process of oxidative stress can induce the expression of the pro-apoptotic protein Bax while inhibiting the expression of the antiapoptotic protein Bcl-2, which leads to an increase in mitochondrial membrane permeability and the release of cytochrome C from mitochondria to the cytoplasm, initiating the caspase cascade and inducing apoptosis [81,82,83,84]. Zhang [85] found that Sal B treatment could reduce the caspase-3 enzyme activity and downregulate the expression of *cytc* and *cas-3* in human cardiovascular endothelial cells (HUVWCs), thereby combating H_2_O_2_-induced apoptosis. Yan et al. [86] also showed that Sal B salt can correct the imbalance of the *bcl-2*/*bax* ratio and reduce the expression of the caspase-3 protein to inhibit the actinomycin D/tumor necrosis factor α-induced apoptosis of human hepatoma cell lines. The results of this study were consistent with the previous studies. After pretreatment with 0.75 or 0.50 g·kg^−1^ Sal B, the apoptosis was significantly improved. The apoptosis-related genes *cas-3*, *-8* and *-9*; *cytc bax*; and *p53* induced by CTX were significantly downregulated. At the same time, the downregulated *bal-2* was significantly restored. This result indicated that CTX could induce hepatocyte apoptosis in Nile tilapia by regulating the expression of key genes in the mitochondrial pathway, while Sal B could alleviate apoptosis.

High levels of ROS can not only directly damage polyunsaturated fatty acids, intracellular proteins, DNA and other biological macromolecules in the biomembrane to induce apoptosis, but can also activate stress-sensitive signaling pathways to initiate apoptosis [81,87]. JNK is a member of the MAPK family, which promotes apoptosis induced by oxidative stress [88,89]. The results of this study showed that the expression levels of *jnk1*, *jnk2*, *ask1*, *p38* and *erk2* in Nile tilapia liver samples were significantly upregulated after CTX exposure, indicating that the CTX treatment induced hepatocyte apoptosis by activating the MAPK signaling pathway. However, pretreatment with 0.75 or 0.50 g·kg^−1^ Sal B significantly inhibited the expression of the genes, suggesting that the effect of Sal B on inhibiting CTX-induced hepatocyte apoptosis may be related to improving the oxidative stress level and inhibiting MAPK signaling. Similar conclusions were also verified in vitro by Zheng et al. [90], who showed that Sal B could reduce intermittent high glucose induced apoptosis of rat insulinoma cells by reducing intracellular the ROS levels and inhibiting *jnk* activation.

## 5. Conclusions

In conclusion, the results of this study indicated that Sal B could significantly resist CTX-induced oxidative injury in Nile tilapia liver samples. In this study, after treatment with 50 mg ·kg^−1^ of CTX, the liver tissue was seriously damaged, the antioxidant capacity of the Nile tilapia was significantly decreased and the levels of autophagy and apoptosis in the hepatocytes were significantly increased. When Sal B pretreatment was administered for 60 days, the level of antioxidant enzymes in the liver tissue was significantly increased and the degree of lipid peroxidation was significantly decreased. At the same time, Sal B significantly improved the growth performance of Nile tilapia, alleviated the morphological lesions of CTX-induced liver tissues and effectively alleviated the autophagy and apoptosis of hepatocytes caused by CTX. In addition, Sal B also significantly regulated the expression of related genes in the antioxidant pathway (Nrf2) and autophagy (AMPK/mTOR) and apoptosis (MAPK) pathways. This suggests that the hepatoprotective effect of Sal B may be achieved through various pathways, including scavenging free radicals and inhibiting hepatocyte apoptosis and autophagy.

## Figures and Tables

**Figure 1 animals-13-00341-f001:**
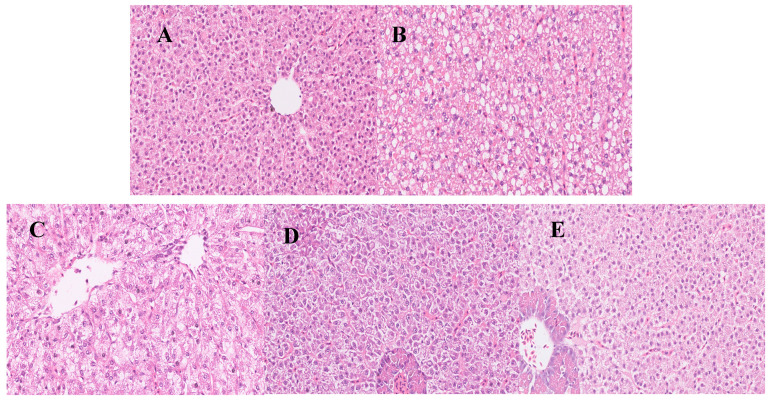
Effects of Sal B on histological changes in Nile tilapia liver sections. The liver sections were stained with haematoxylin and eosin. All sections were photographed with a microscope (400× magnification): (**A**) normal saline alone; (**B**) CTX alone; (**C**) CTX plus 0.25 g·kg^−1^ Sal B; (**D**) CTX plus 0.50 g·kg^−1^ Sal B; (**E**) CTX plus 0.75 g·kg^−1^ Sal B.

**Figure 2 animals-13-00341-f002:**
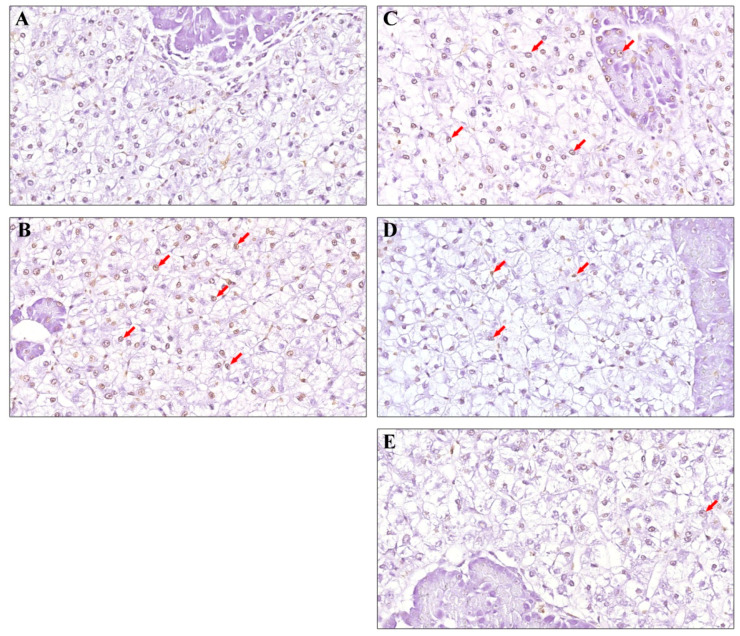
Effect of Sal B on apoptosis of hepatocytes of Nile tilapia. All sections were photographed with a microscope (400× magnification): (**A**) normal saline alone; (**B**) CTX alone; (**C**) CTX plus 0.25 g·kg^−1^ Sal B; (**D**) CTX plus 0.50 g·kg^−1^ Sal B; (**E**) CTX plus 0.75 g·kg^−1^ Sal B. Apoptotic nuclei were stained brown and the red arrows show apoptotic cells.

**Figure 3 animals-13-00341-f003:**
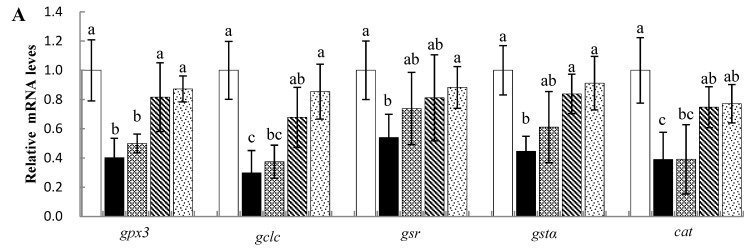
Effects of Sal B on antioxidant and Nrf2-pathway-related gene mRNA levels in liver samples of Nile tilapia: (**A**) expression of antioxidant-related genes; (**B**) expression of Nrf2-pathway-related genes. The values are means ± SE (*n* = 8). ^a,b,c^ Different superscript letters for each parameter indicate significant differences (*p* < 0.05).

**Figure 4 animals-13-00341-f004:**
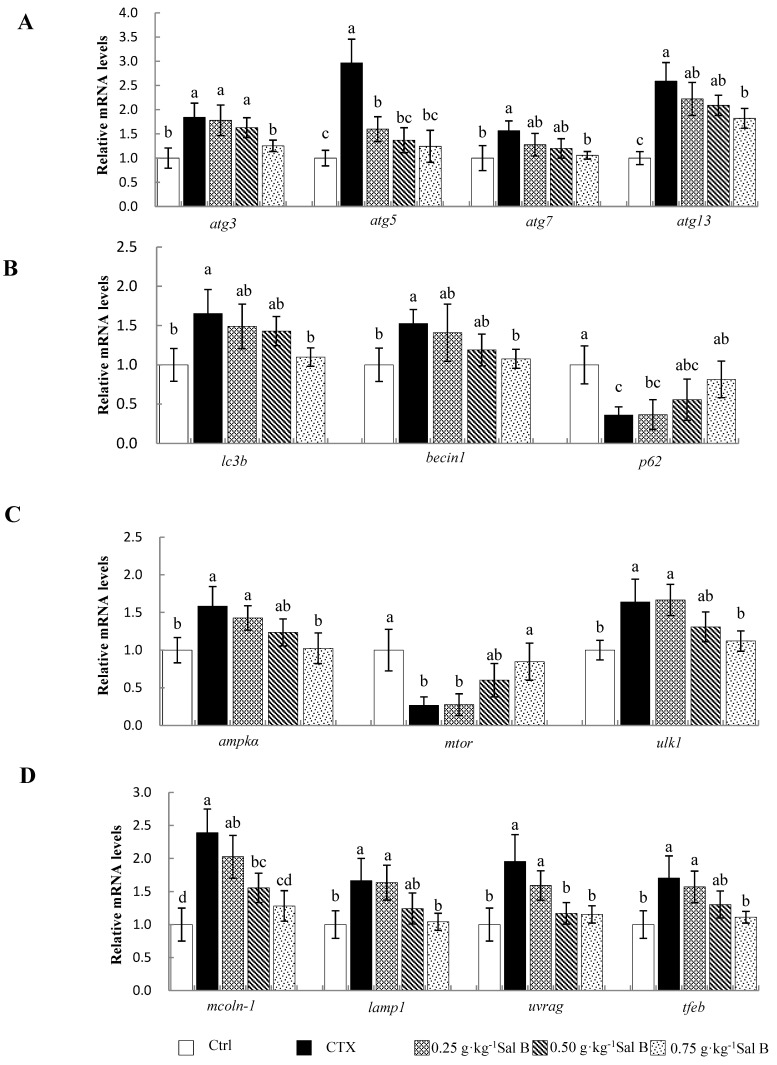
Effects of Sal B on autophagy and AMPK/mTOR-pathway-related gene mRNA levels in liver samples of Nile tilapia: (**A**,**B**) expression of autophagy-related genes; (**C**,**D**) expression of AMPK/mTOR-pathway-related genes. The values are means ± SE (*n* = 8). ^a,b,c^ Different superscript letters for each parameter indicate significant differences (*p* < 0.05).

**Figure 5 animals-13-00341-f005:**
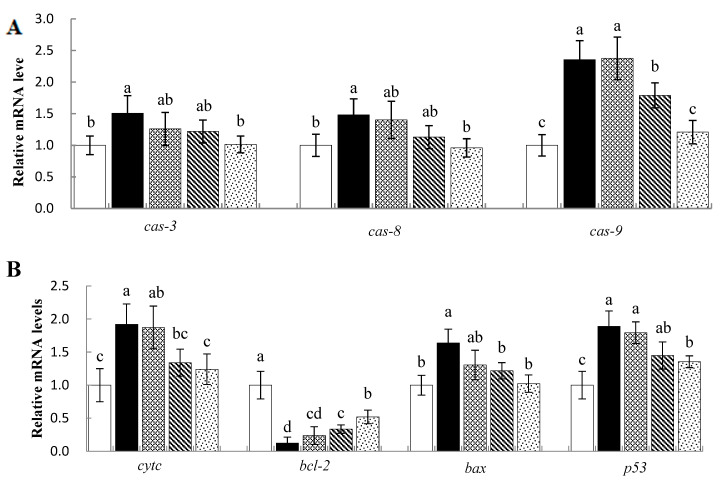
Effects of Sal B on apoptosis and MAPK-pathway-related genes mRNA levels in liver of Nile tilapia. (**A**,**B**), expression of apoptosis-related genes; (**C**), expression of MAPK-pathway-related genes. The values are means ± SE (*n* = 8). ^a,b,c^ Different superscript letters for each parameter indicate significant differences (*p* < 0.05).

**Table 1 animals-13-00341-t001:** Compositions and nutrient levels of the experimental diets.

Component/g·kg^−1^	Control	0.25 g·kg^−1^ Sal B	0.50 g·kg^−1^ Sal B	0.75 g·kg^−1^ Sal B
Fish meal	60.0	60.0	60.0	60.0
Rapeseed meal	270.0	270.0	270.0	270.0
Flour	174	173.75	173.50	173.25
Sal B	0	0.25	0.50	0.75
Rice bran meal	100.0	100.0	100.0	100.0
Cottonseed meal	100.0	100.0	100.0	100.0
Soyabean meal	240	240	240	240
Phospholipid	10	10	10	10
Soybean oil	10	10	10	10
Monocalcium phosphate	15	15	15	15
Choline chloride	1	1	1	1
Vitamin premix ^1^	10	10	10	10
Mineral premix ^2^	10	10	10	10
Total	1000	1000	1000	1000
Approximate composion				
Crude protein/%	31.6	31.6	31.6	31.6
Crude lipid/%	4.6	4.6	4.6	4.6
Ash/%	6.4	6.4	6.4	6.4

Note: ^1^—Vitamins (mg/ kg diet): VA 10 mg, VB_1_ 50 mg, VB_2_ 200 mg, VB_3_ 500 mg, VB_6_ 50 mg, VB_7_ 5 mg, VB_11_ 15 mg, VB_12_ 0.1 mg, VC 1 000 mg, VD 0.05 mg, VE 400 mg, VK 40 mg, inositol 2000 mg, choline 5000 mg; ^2^—minerals (mg/ kg diet): Fe (FeSO_4_-7H_2_O) 372 mg, Cu (CuSO_4_·5H_2_O) 25 mg, Zn (ZnSO_4_·7H_2_O) 120 mg, Mn (MnSO_4_-H_2_O) 5 mg, Mg (MgSO_4_) 2475 mg, NaCl l875 mg, KH_2_PO_4_ 1000 mg, Ca (H_2_PO_4_)_2_ 2500 mg.

**Table 2 animals-13-00341-t002:** The primer sequences used in the present study.

Gene	Primer Sequence (5′-3′)	GenBank Number/References
*nrf2*	F: CTGCCGTAAACGCAAGATGG	XM_003447296.4
	R: ATCCGTTGACTGCTGAAGGG	
*keap1*	F: CTTCGCCATCATGAACGAGC	XM_003447926.3
	R: CACCAACTCCATACCGCACT	
*ucp2*	F: CAGGGATCGTTACTCGGCTC	XM_019363847.2
	R: CCGTTGTATCTCCTCTCGCC	
*ho-1*	F: CTTGCCCGTGTGGAATCACT	XM_013270165.2
	R: AGATCACCGAGGTAGCGAGT	
*gpx3*	F: CGATGTGGCCCGTGTTACC	XM_005467951.3
	R: CAGCCTTGCCTGCGTAGT	
*gclc*	F: ATCGAGGGAACTCCAGGTCA	XM_003441123.5
	R: GGAGTTGGTCGGTACTCTGG	
*gsr*	F: TACGCTGTCGGGGATGTTTG	XM_003445184.5
	R: AATGGCCTCGTCCTCTGTGA	
*gstα*	F: TAATGGGAGAGGGAAGATGG	NM_001279635.1 [42]
	R: CTCTGCGATGTAATTCAGGA	
*cat*	F: ATGGCCACCGTCACATGAAT	XM_019361816.2
	R: CAAACGGGTTGAACCGGAAC	
*jnk1*	F: CAAGCCCGTGACCTCCTATC	XM_025909605.1
	R: TCCATTCCTCCACTGTGTGC	
*jnk2*	F: GCACGGGATTTGCTTTCCAA	XM_005467784.4
	R: TGTGCTCTCTCTCTTCCAGC	
*ask1*	F: TCTCCCACGAATCCCACAATG	XM_003446319.5
	R: GCTCAGTTCGCAGCTTCAGT	
*p38*	F: GGATGACCACGTCCAGTTCC	XM_003441528.5
	R: TTTCATCATCCGTGTGCCGT	
*erk2*	F: CGAAGGGCTACACCAAGTCC	XM_003444474.5
	R: GATGATGCAGTTGAGATCCTCCT	
*cas3*	F: CAGACAGCGAAACTGATGGTG	NM_001282894.1
	R: CACCTTGTGGTTCACTCGGG	
*cas8*	F: AGGGTGTAGTTTTGGGAGCTG	XM_019348156.2
	R: GGGGATCGTCAATGGTGAAGTA	
*cas9*	F: AGACGGACGCTATTCCGATG	XM_025901776.1
	R: CGCCAAGAAACATAACCTGGG	
*cytc*	F: GACGCCAACAAGAGCAAAGG	XM_005473697.4
	R: AATGAGGTCTTGGCGCTCTC	
*Bcl2*	F: AGACTGTACCAGCCGGACTT	XM_003437902.5
	R: GTGCCCCCAAACTCGAAGAA	
*bax*	F: TGGCAATAAAGCAGTGACGAG	XM_019357746.2
	R: AGGCCACTCTCATAAAAACCTC	
*p53*	F: GGCAATCAGAGGGCTCAGTA	XM_025905405.1
	R: GTGAGGATAGGTCTGCGGTT	
*ampkα*	F: GTGGGGTGATTCTCTACGCC	NM_001319868.1
	R: GGCTCTTTTCATCGGGTCCA	
*mtor*	F: AGCGACAGTGAGGTTGACAG	XM_003449131.5
	R: GGACAGTGAAATGGAGCGGA	
*ulk1*	F: GGTCCAGAATTACCAGCGCA	XM_019361142.2
	R: CCCATAGGGTGGTGAAGGTC	
*mcoln-1*	F: CTGCTGTGTGGCTGTCATCT	XM_005470179.4
	R: GACTGGTCTCCTGCATCTCTG	
*lamp1*	F: CTGGTTTTGACACAGACGCAA	XM_003445782.5
	R: ATGTAGGAGTAGCCCAGCGT	
*uvrag*	F: ATCACCCACACACCTGTCATC	XM_003458979.5
	R: TGTTCAGCGGTTTTACGTCCT	
*tfeb*	F: GCACCTACCCCATAACCAGG	XM_019358786.2
	R: GCAAAGTCAAAGTGCTGGGAG	
*atg3*	F: CGACTCTGGCTCTTTGGATATGA	XM_003454721.5
	R: CCTTCCGCCACAGTCTCAAT	
*atg5*	F: GGATGGGCTTGCAGAACGATA	XM_003450274.5
	R: AAGGGTGTATGCGTTGCCT	
*atg7*	F: TCTCTCAGACCACTCTGTCCC	XM_025907253.1
	R: AGCAGCATTCACCACTAGCTT	
*atg13*	F: CGATGATGATGGCTTGTCGC	XM_005460641.4
	R: AAACGCAGCAAATGGCAGG	
*lc3b*	F: GCACCCCAACAAAATACCTGT	XM_003439438.5
	R: CCTGGTTGGAGTTTAGCTGGA	
*beclin1*	F: ACCATCAACAACTTCCGCCT	XM_005471281.3
	R: TCTGAAAGTGCAGCCCCATT	
*p62*	F: CCCTTCTAAACCTGCTGCTGA	XM_005463795.4
	R: TCACCTTGGTCCGTTGGC	
*β-actin*	F: CCTGAGCGTAAATACTCCGTCTG	KJ126772.1 [43]
	R: AAGCACTTGCGGTGGACGAT	

**Table 3 animals-13-00341-t003:** Effects of Sal B on growth indexes of Nile tilapia.

Group	Initial Body Weight/g	Final Weight/g	RWR/%	SGR/(%·d^−1^)	FCR
Control	30.04 ± 1.09	89.12 ± 3.03 ^a^	196.67 ± 5.71 ^a^	1.81 ± 0.05 ^a^	2.96 ± 0.17 ^a^
CTX	30. 08 ± 1.13	92.88 ± 2.69 ^a^	208.78 ± 6.05 ^a^	1.88 ± 0.07 ^a^	2.78 ± 0.11 ^a^
0.25 g·kg^−1^ Sal B	30.00 ± 0.92	97.80 ± 3.21 ^ab^	226.00 ± 6.39 ^ab^	1.97 ± 0.12 ^b^	2.58 ± 0.12 ^ab^
0.50 g·kg^−1^ Sal B	29.75 ± 1.05	98.14 ± 3.08 ^b^	229.88 ± 6.39 ^b^	1.99 ± 0.08 ^b^	2.55 ± 0.10 ^b^
0.75 g·kg^−1^ Sal B	29.91 ± 0.89	100.24 ± 3.57 ^b^	235.14 ± 8.01 ^b^	2.02 ± 0.13 ^b^	2.48 ± 0.13 ^b^

Note: Values are expressed as the means ± SE (*n* = 8). ^a,b^ Different superscript letters for each parameter indicate significant differences (*p* < 0.05).

**Table 4 animals-13-00341-t004:** Effects of Sal B on biochemical indexes in serum and liver samples of Nile tilapia.

Tissues	Parameters	Control	CTX	0.25 g·kg^−1^ Sal B	0.50 g·kg^−1^ Sal B	0.75 g·kg^−1^ Sal B
Serum	GPT/IU·L^−1^	3.24 ± 0.48 ^b^	8.12 ± 1.64 ^a^	6.11 ± 0.82 ^a^	4.77 ± 0.63 ^ab^	3.17 ± 0.36 ^b^
	GOT/IU·L^−1^	3.15 ± 0.22 ^c^	9.78 ± 0.77 ^a^	7.46 ± 0.80 ^b^	4.17 ± 0.53 ^c^	4.00 ± 0.56 ^c^
	TP/g·L^−1^	22.12 ± 0.94 ^a^	15.77 ± 0.63 ^c^	17.97 ± 1.12 ^bc^	18.54 ± 1.24 ^bc^	21.17 ± 1.73 ^ab^
	Alb/g·L^−1^	16.08 ± 0.89 ^a^	12.31 ± 0.74 ^c^	12.97 ± 0.44 ^bc^	14.66 ± 0.94 ^ab^	15.05 ± 0.88 ^ab^
	T-AOC/mmol·L^−1^	2.18 ± 0.18 ^a^	0.76 ± 0.02 ^c^	0.76 ± 0.05 ^c^	1.08 ± 0.13 ^c^	1.54 ± 0.15 ^b^
Liver	8-OHdG/ng·g prot^−1^	274.13 ± 9.12 ^c^	332.41 ± 12.10 ^a^	328.98 ± 10.34 ^ab^	303.49 ± 15.64 ^abc^	295.99 ± 14.24 ^bc^
	NO/μM·mg prot^−1^	5.92 ± 0.36 ^a^	3.54 ± 0.57 ^b^	3.81 ± 0.33 ^b^	4.74 ± 0.58 ^ab^	5.08 ± 0.46 ^a^
	T-AOC/μmol·g prot^−1^	79.91 ± 5.70 ^a^	49.52 ± 3.86 ^c^	65.04 ± 3.50 ^b^	72.02 ± 6.42 ^ab^	75.24 ± 5.34 ^ab^
	SOD/U·mg prot^−1^	92.54 ± 8.39 ^a^	69.00 ± 3.05 ^c^	70.90 ± 6.95 ^bc^	87.83 ± 8.48 ^ab^	88.11 ± 4.03 ^ab^
	CAT/U·mg prot^−1^	800.58 ± 17.67 ^a^	641.55 ± 24.21 ^c^	644.84 ± 41.62 ^c^	668.84 ± 49.38 ^bc^	770.36 ± 57.97 ^ab^
	MDA/nmol·mg port^−1^	0.37 ± 0.02 ^b^	0.62 ± 0.03 ^a^	0.58 ± 0.07 ^a^	0.52 ± 0.03 ^a^	0.40 ± 0.04 ^b^
	GSH/μmol ·g prot^−1^	40.06 ± 9.88 ^a^	13.94 ± 1.76 ^c^	15.42 ± 8.16 ^bc^	26.81 ± 9.76 ^abc^	38.98 ± 9.74 ^ab^

Note: Values are expressed as the means ± SE (*n* = 8). ^a,b,c^ Different superscript letters for each parameter indicate significant differences (*p* < 0.05).

## Data Availability

Data sharing not applicable.

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
