# Peer review of "Salvianolic Acid B Regulates Oxidative Stress, Autophagy and Apoptosis against Cyclophosphamide-Induced Hepatic Injury in Nile Tilapia (Oreochromis niloticus)"

_animals, 2023, doi:10.3390/ani13030341_

Round 1
Reviewer 1 Report
This study reported a protective effect of Sal B, a plant extract, on tilapia liver injury induced by cyclophosphamide. The methods are appropriate, and the results are credible. However, there are some error in the current manuscript, which need to revise.
Major comments
1. line 42, please delete the word “material” and “water”.
2. Line 67,studies by zhang, should be study by zhang.
3. Line 79 “a good protective effects” should be” a good protective effect”.
4. In the third paragraph of the introduction, some references about Chinese herbal medicine against cyclophosphamide induced liver injury should be added.
5. In the treatments, why authors suggested that 0.25 g·kg-1 is low dose group and 0.75 g·kg-1 is high dose group.
6.line 114, How to determine the concentration of anesthetic? Please add references.
7. Provide some references in the biochemical indicators.
8.” Determination of gene mRNA level” change into “Determination of mRNA level”.
9. Please provide qPCR kits Cargo number
10. Line 189 and 190,what means of ‘No captial letters………”
11. In table, are there references?
12. The units of biochemical indicators in Table 4 need to be checked and verified
13. Line251, “no small letters…….. ”should be “no same letters….”
14. why author measured one oxidative stress index (T-AOC) in serum?
15. in figure3B, please check the keap1 gene
16. Suggest delete the figure 5C, it is suitable for protein expression only.
17. Line354 “In recent years, many experiments have confirmed the antioxidant activity of Sal B directly or indirectly” add references
18. Line365 was inhibited should be were inhibited.
19. In the discussion section, there is too little discussion about drugs to protect liver damage, please improve it.
20. Line 404 was should be were.
21. Histological changes were not mentioned in the discussion
22. figure is not enough clearly.
23. Some grammar errors need to revise.
Reviewer 2 Report
This manuscript focuses on an actual and interesting topic and fits the scopes of this journal.
1. There is not a good introduction of the topic, only an enumeration of these key points is presented.
Ex:Line 76-93: Salvianolic acid B (Sal B) introduction.
2. Many important things are omitted in materials and methods. For example, Elisa methods , number of replicates, fish maintenance, survival rates, etc. Lack of such essential information hinder readers to clearly understand experiments and to evaluate data presented.
3. Overall, their experiments were carried out according to standard and routine procedures, and results did not deviate significantly from general expectations.
4. Authors should add reference(s) at the end of following sentence:
Ex: Line 46-51:In recent years, with the continuous development of aquaculture intensification, nutrient deficiencies, antibiotic abuse and drug residues are prevalent, and all these undesirable factors can lead to hepatic injury in aquatic animals. Liver injury or lesions often lead to metabolic disorders and reduced disease resistance in fish, which can easily lead to outbreaks of secondary infectious diseases and seriously restrict the healthy development of aquaculture.
5. In addition, the list of references is not in our style. It is close but not completely correct.
Ex: Line 474: An, J.J., Shi, K.J., Wei, W., Hua, F.Y., Ci, Y.L., Jiang, Q., Li, F., Wu, P., Hui, K.Y., Yang, Y., 2017. The ROS/jnk/ATF2 pathway mediates selenite- induced leukemia NB4 cell cycle arrest and apoptosis in vitro and in vivo. Cell death and disease 4, 1-10.
Cell Death And Disease
3. Figure 3.B, Expression of Nrf2 pathway-related genes.
Line 259: Keap1 gene CTX group set errors .
Reviewer 3 Report
1. Since the authors already mentioned the downside of aquaculture practice to the liver function such as “In recent years… aquatic animals (Line 46-49) and high-fat diets problem (Line 52)”, why the experiment was not conducted using these stressors but cyclophosphamide (CTX)? And why CTX is relevant to aquaculture practice?
2. Please provide the scientific name of tilapia used in this study.
3. Line 114, the authors stated that “Fasting for 72 h was used to induce liver damage.”. Was fasting or injection of CTX used for liver damage induction?
4. Were the fish fed with experimental diets before and after CTX injection?
Round 2
Reviewer 3 Report
The authors have addressed all the questions raised and the manuscirpt is now ready to be accepted.